Towards a unified generic framework to define and observe contacts between livestock and wildlife: a systematic review

Bacigalupo Sonny A. sbacigalupo@rvc.ac.uk 1
Dixon Linda K. 2
Gubbins Simon 2
Kucharski Adam J. 3
Drewe Julian A. 1
1 Royal Veterinary College , London , United Kingdom
2 The Pirbright Institute , Woking, Surrey , United Kingdom
3 London School of Hygiene & Tropical Medicine, University of London , London , United Kingdom
Aly Sharif
Electronic publication date: 2020 Oct 26
Publication date: 2020
Volume: 8
Electronic Location ID: e10221
Received 2020 May 15; Accepted 2020 Sep 29
Copyright: ©2020 Bacigalupo et al.
Copyright year: 2020
Copyright holder: Bacigalupo et al.
License: This is an open access article distributed under the terms of the Creative Commons Attribution License, which permits unrestricted use, distribution, reproduction and adaptation in any medium and for any purpose provided that it is properly attributed. For attribution, the original author(s), title, publication source (PeerJ) and either DOI or URL of the article must be cited.
License URL: https://creativecommons.org/licenses/by/4.0/

Keywords: Wildlife-livestock, Disease transmission, Contact, Interaction, Interface, Framework, Definition, Methods

Funding: Bloomsbury PhD studentship London Interdisciplinary Doctoral Programme BB/M009513/1 Sonny A.J. Bacigalupo is supported by a Bloomsbury PhD studentship and the Royal Veterinary College through the London Interdisciplinary Doctoral Programme (BBSRC project number BB/M009513/1). The funders had no role in study design, data collection and analysis, decision to publish, or preparation of the manuscript.

==============================
Wild animals are the source of many pathogens of livestock and humans. Concerns about the potential transmission of economically important and zoonotic diseases from wildlife have led to increased surveillance at the livestock-wildlife interface. Knowledge of the types, frequency and duration of contacts between livestock and wildlife is necessary to identify risk factors for disease transmission and to design possible mitigation strategies. Observing the behaviour of many wildlife species is challenging due to their cryptic nature and avoidance of humans, meaning there are relatively few studies in this area. Further, a consensus on the definition of what constitutes a ‘contact’ between wildlife and livestock is lacking. A systematic review was conducted to investigate which livestock-wildlife contacts have been studied and why, as well as the methods used to observe each species. Over 30,000 publications were screened, of which 122 fulfilled specific criteria for inclusion in the analysis. The majority of studies examined cattle contacts with badgers or with deer; studies involving wild pig contacts with cattle or with domestic pigs were the next most frequent. There was a range of observational methods including motion-activated cameras and global positioning system collars. As a result of the wide variation and lack of consensus in the definitions of direct and indirect contacts, we developed a unified framework to define livestock-wildlife contacts that is sufficiently flexible to be applied to most wildlife and livestock species for non-vector-borne diseases. We hope this framework will help standardise the collection and reporting of contact data; a valuable step towards being able to compare the efficacy of wildlife-livestock observation methods. In doing so, it may aid the development of better disease transmission models and improve the design and effectiveness of interventions to reduce or prevent disease transmission.

Introduction

The interface where livestock and wildlife may come into contact with each other is an area of growing scientific interest, particularly as wildlife can act as a ‘reservoir’ for diseases of livestock (Wiethoelter et al., 2015). Disease transmission between livestock and wildlife can have marked economic impact, such as African swine fever outbreaks in domestic pigs and wild boar (Sus scrofa) in Europe and Asia (Dixon et al., 2019), where the loss of 12–20% of the global pig herd in 2019 led to a 10% increase in the food price index of pork (Pitts & Whitnall, 2019). The impact of disease transmission on wildlife can be seen in the loss of around half the global saiga (Saiga tatarica) antelope population in 2015 to Pasteurella multocida, a pathogen harboured by livestock (Fereidouni et al., 2019). Contact between wildlife and livestock may also lead to conflict between humans and wildlife, with compensation for large carnivore predation and other damage costing 28.5 million euros annually in Europe (Bautista et al., 2019). The proximity of agricultural land to wildlife habitats is a key factor in human-wildlife conflicts and in the spill-over of pathogens from wildlife to livestock and humans (Jones et al., 2013). The emergence of diseases from wildlife that infect humans via livestock intermediaries, such as bat-borne Hendra virus (affecting humans via horses) and Nipah virus (affecting humans via pigs) (Field et al., 2001), further highlight the importance of contacts between wildlife, livestock and people. These contacts are seldom recorded, however, because many wildlife species are cryptic and therefore difficult to observe, capture and sample.

Observing wildlife-livestock contacts is becoming easier with advances in remote technologies such as motion-activated cameras, global positioning system (GPS) collars and proximity loggers (Böhm, Hutchings & White, 2009; Drewe et al., 2013; Barasona et al., 2014). These methods are usually (but not always) used to monitor one species at a time. They are not standardised, however, meaning there are many variations in monitoring protocols, often depending on basic practicalities such as battery life, people-hours, cost and the aims of the study. The methods used to monitor livestock-wildlife contacts may influence (or be influenced by) the kind of contact to be monitored, the context of the study and what the data will be used for.

Livestock-wildlife contact data is needed to inform the simulation and modelling of diseases that have multiple host species, but information on the types of contact needed for transmission and the rates at which these occur is lacking (Craft, 2015). Knowledge of livestock-wildlife contact data can be used to identify risk factors and predict where these contacts are more or less likely to occur, for example predicting the likelihood of badger (Meles meles) visits to cattle farms in the context of bovine tuberculosis transmission (Robertson et al., 2019). It could also be used to implement and improve mitigation strategies to prevent unwanted livestock-wildlife contacts. To mitigate wolf (Canis lupus) predation on sheep, for example, the effectiveness of prevention programs needs to be evaluated in ways that do not depend on livestock attacks alone, using methods such as GPS monitoring of wolf movements around sheep farm bio-fences (Bautista et al., 2019; Ausband et al., 2013). Similarly, the effectiveness of measures taken to prevent disease transmission can also be evaluated such as by comparing deer-cattle contact rates between farms with and without deer fences installed (Lavelle et al., 2015; Lavelle et al., 2016; Wilber et al., 2019). Knowledge of livestock-wildlife contacts can be used in these contexts to limit the economic loss associated with disease and predation. Given these multiple ways of gathering and using livestock-wildlife contact data, the definition of what constitutes a relevant contact will vary depending on the aim of the study.

In the context of disease transmission, defining contact is challenging and while types of contact are often broadly grouped into being ‘direct’ or ‘indirect’, there are no standardised definitions (Eames et al., 2015). Direct contacts are usually thought of as representing physical contact or being in close proximity over a short period of time, and so may include fighting, mating between feral and domestic animals of the same species, or being face-to-face or nose-to-nose. Indirect contacts are more difficult to define due to issues of long-distance aerosol transmission, environmental persistence of pathogens in spores and fomites, and intermediate insect vectors (Craft, 2015). Other ecological definitions of livestock-wildlife contacts could also include avoidance behaviour or competition for resources between species. This variation in definitions means it is difficult to make meaningful comparisons between studies and to apply findings from one study to different contexts. Therefore, a standardised generic template for defining livestock-wildlife contacts would be useful.

The aim of this study was to systematically review the reasons for, and observational methods used in, studies investigating livestock-wildlife contacts, and to propose a generalised framework for defining contacts between livestock and wildlife.

Methods

Literature search and data extraction

We defined livestock as ‘farmed domesticated mammals’ (FAO, 2020), wild animals as ‘free-ranging non-domesticated mammals’, and contact as ‘activity implying an interaction or association between species including the shared use of resources such as farmland’. The terms interaction and contact were used synonymously within the literature, but contact is used here for consistency. The systematic review question was “Which methods have been used to assess the frequency of, and types of, contacts between wild animals and livestock or livestock farms worldwide?”.

Search terms for wildlife, livestock and type of contact were combined by the Boolean operators ‘OR’ and ‘AND’ to identify publications that investigated contact between any wild and domestic mammal (Table S1). Search terms were based on common species names, and generic terms such as ‘feral’, ‘wildlife’, ‘livestock’ and ‘farm’. Searches were conducted in CAB Abstracts, Scopus and Pubmed. CAB Abstracts is a comprehensive database of life science research with broad coverage of veterinary literature in particular, and Scopus has a broad coverage of interdisciplinary journals (Grindlay, Brennan & Dean, 2012; Aghaei Chadegani et al., 2013).

Search results were consolidated into Microsoft Excel and duplicates were identified and removed using queries followed by manual inspection. Titles, abstracts and full texts of the retrieved publications were evaluated by SAB against pre-specified exclusion and inclusion criteria (Table 1). Any papers for which the criteria were not clear were also evaluated by JAD. In all such cases both authors agreed on the final decision. We wished to capture publications that collected, used or analysed data to investigate direct or indirect contacts between farmed livestock and terrestrial wild mammals whose adult bodyweight is typically >5 kg. Specifically, publications were included if they attempted to quantify, characterise, or identify risk factors for livestock-wildlife contacts. Only articles in English and those accessible to researchers were included. All reasonable efforts were made to access papers that passed abstract screening. We excluded studies in which predation events were the sole indicator of livestock-wildlife contacts, and studies of wild animals that were not free-living, were tamed or were relocated for the purpose of the study. Publications until 11 November 2019 were included, and no time restrictions were applied to the start of the search. Working definitions of direct and indirect contact were developed before performing the literature search and used to avoid ambiguity when evaluating publications for inclusion.

Table 1 Exclusion and inclusion criteria to select studies for the systematic review of livestock-wildlife contact.

Exclusion Criteria	
1. Study does not involve a wild mammal species where adults are typically heavier than 5 kg.	
2. Study does not involve a farmed mammal species where adults are typically heavier than 5 kg, or farmland associated with such livestock.	
3. Study does not attempt to collect, use or analyse data to investigate contacts between wild animals and livestock or livestock farms.	
4. Study does not attempt to collect, use or analyse data to establish at least one of the following: characterisation of, the nature of, frequency of, or risk factors for, contacts between wildlife and livestock.	
5. Full text not available in English.	
6. Full text not accessible to reviewers.	
7. The method of recording livestock-wildlife contacts relies solely on predation events where the only observations are livestock kills or scat analysis	
8. Wild animals were non-free-living, pre-tamed or relocated for the purpose of the study.	
Inclusion Criteria	
The study aims to collect, use, or analyse data to establish at least one of the following:	
1. A quantifiable measure of direct contact between wildlife and livestock, where direct contact is defined as physical contact between at least one wild animal and one farm animal.	
2. A quantifiable measure of indirect contact between wildlife and livestock, where indirect contact is defined as contact between at least one wild animal and a resource used by at least one farm animal including, but not limited to, food, water and space	
3. Characterise and establish the type of, or risk factors for, direct or indirect contact between wildlife and livestock, as defined above.	

Direct contact was provisionally defined prior to reviewing the papers as physical contact between at least one wild animal and one farm animal. Indirect contact was provisionally defined as contact between at least one wild animal and a resource used by at least one farm animal including, but not limited to, food, water and space. Therefore, studies that investigated wildlife and livestock shared resource use, but did not explicitly investigate contacts, were included. These definitions were used throughout the process of identifying and analysing the papers in this review. Study data was extracted and livestock and wildlife species, observation methods and definitions were categorised. Where available, the power of each study, defined as the likelihood of detecting contacts, was recorded. Themes that emerged during data extraction were grouped into seven broad study themes, namely behavioural, competition, conservation, disease, human-wildlife conflict, methods papers and wildlife management (Fig. S1). Where studies had more than one theme, themes were subjectively allocated as dominant (primary) or secondary based on the aims of the study. Results were visualised and plotted using R (version 3.6.3 (R Core Team, 2020)) and R packages listed in Table S2.

Development of a generic unified framework

Following categorisation of definitions, a generic unified framework was developed by grouping and identifying commonalities in definitions of ‘direct’ and ‘indirect’ contact, namely relating to space and time. The spatial and temporal limits separating relevant contacts from inconsequential contacts and non-contact events were identified for each study, and a framework was developed based on defining contacts in relation to both space and time. Using this framework, relevant contacts were defined using the parameters of critical space (SC) and critical time (TC). We defined SC as the critical space (distance or area) between animals below which a contact relevant to the study is considered to have occurred, and TC as the critical time window within which a relevant contact is considered to have occurred.

Results

During data categorisation and analyses, many publications were categorised into more than one group due to studying multiple species, using multiple detection methods and having multiple themes, and therefore the number of studies exceed 122 (100%) in several instances reported below.

Search results, quality appraisal and themes

A total of 43,032 papers were identified by the search terms across all three databases, of which 30,080 were unique results. After screening using the exclusion and inclusion criteria in Table 1, 122 publications remained in the final analysis (Fig. 1). Publication date ranged from 1980 to 2019, with 117 (96%) published in the last 20 years (Fig. 2). Studies conducted in Europe, North America and Africa made up 89% of the results (Table S3) with the USA and UK producing the most publications (21% and 18%, respectively).

Figure 1 Flow chart documenting literature retrieval and criteria used to select articles for inclusion in the systematic review of direct and indirect contacts between wildlife and livestock.

Search categories (contact term, livestock and wildlife) were combined by the Boolean operator ‘AND’ to identify publications containing all three terms. Databases were searched up to 11 November 2019 with no historic limit.

Figure 2 Distribution of the publication year of 122 publications included in the systematic review.

Publication date ranged from 1980 to 2019, with 117 (96%) published in the last 20 years.

Low study power was mentioned briefly in only 11 (9%) publications and statistical power calculations were not performed. The level of uncertainty was acknowledged in 64 (53%) publications.

Disease was the dominant theme and featured in 80 of 122 studies (66%), followed by human-wildlife conflict (22/122; 18%), competition between wildlife and livestock (17/122; 14%), conservation (16/122; 13%), wildlife management (11/122; 9%), behavioural studies (3/122; 2%) and methods validation (2/122; 2%) (Fig. S1). Within the disease-themed papers, Mycobacterium bovis was the most studied pathogen (49/80; 61%) followed by foot-and-mouth disease virus (8/80; 10%) (Tables S4 and S5). Wildlife-cattle contacts were the focus of 98 of the 122 studies (80%) and a further 22 studies (18%) focussed on sheep, pigs, farmed deer and camelids. The most studied wildlife species were deer (30/122; 25%), wild pigs [including wild boar] (26/122; 21%) and badgers (25/122; 20%: Figs. S2 and S3). The wildlife species were not specified in 11 papers, some of which studied wild ungulates competing for livestock grazing (Mizutani, Kadohira & Phiri, 2012; Sitters et al., 2009; Crawford et al., 2019), others that concerned wildlife as hosts of cattle diseases such as bovine tuberculosis (Munyeme et al., 2010; Witmer et al., 2010; Katale et al., 2013) and foot-and-mouth disease (Brahmbhatt et al., 2012; Molla et al., 2013), and the remainder that were completely unspecified.

Methods used to observe livestock-wildlife contacts

Methods that monitored both livestock and wildlife species were used in 88 publications (72%) whereas 34 studies (28%) monitored wildlife only. Camera trapping was the most frequent method of monitoring wildlife (37 studies, 31%), and was most prominently used in badgers, deer and wild pigs (Fig. 3). GPS collars were the second most used method to monitor wildlife (29 studies, 24%), and while they were also used predominantly on badgers, deer and wild pigs, they were used proportionally more than cameras to monitor predators such as big cats and wolves and large herbivores such as buffalo, wild horses and elephants. Other methods used to monitor wildlife were direct visualisation (21; 17%), farmer questioning (20; 16%), radio-transmitters (17; 14%), activity signs (15; 12%) and proximity loggers (7; 6%). Some studies utilised more than one observation method, and therefore the numbers of studies exceed 122 (100%) Studies that monitored livestock tended to use the same methods as for wildlife, although 10 studies dedicated fewer resources to monitor livestock; for example (Pruvot et al., 2014) used GPS collars to monitor wild deer and farmer questioning to monitor cattle behaviour. Studies that did not monitor livestock tended to infer wildlife-livestock contact from monitoring only the activities of wildlife on or around livestock holdings, such as on pasture, in buildings and the shared use of resources such as livestock feed.

Figure 3 Observation methods used to monitor wildlife.

Data from 122 publications included in the systematic review. The size and shade of circles indicate the number of studies in each category. Many publications used more than one method to monitor contacts, and therefore the number of studies exceeds 122 (100%) for some groups.

A variety of methods were used to observe different types of contact data (Fig. S4). Methods such as GPS collars and radio-tracking (telemetry) were used to collect the locations of wildlife (e.g., Barasona et al., 2014; Raizman et al., 2013; Cooper et al., 2008), whereas proximity loggers were used to detect close proximity contacts between livestock and wildlife or with postulated high-risk disease transmission areas such as badger latrines (e.g., Drewe et al., 2013). Camera traps and direct visualisation were used to observe behavioural activity, such as nose-to-nose contacts between cattle and badgers (Tolhurst et al., 2009), foxes taking piglets from farrowing huts (Fleming et al., 2016) and wild boar eating from cattle troughs (Kukielka et al., 2013). Some methods were used to detect the presence of wild animals on farms or on pasture only, such as surveys of activity signs to detect wild boar rooting on sheep pasture (Guillermo Bueno et al., 2010) and GPS collars to demonstrate the avoidance of livestock pasture by lions (Oriol-Cotterill et al., 2015). Thirty studies combined more than one method to monitor wildlife, such as (Wyckoff et al., 2012) which combined activity signs, GPS collar data and camera traps to monitor feral swine activity at and around domestic pig pens. The majority of studies, however, used only one method and were able to collect information about the type of contact defined by the study.

Definitions of direct and indirect contacts

Definitions for both direct contact and indirect contact were provided by 27 studies, with a further four defining direct contact only and 54 defining indirect contact only (Tables 2 and 3). Definitions of direct contact tended to focus on the spatial distance between wildlife and livestock at one point in time (Table 2). Definitions of indirect contact tended to focus on the use of space or resources by wildlife in a location previously or subsequently occupied by livestock, within a certain time frame (Table 3). There were some variations to these trends: two studies specified a time frame longer than one time point to define direct contact (Lavelle et al., 2016; Cooper et al., 2010). The amount of time was usually determined by the context of the study, such as the survival time of a specified pathogen in the environment, known as the critical time window of a contact (Cowie et al., 2016). Contacts were also defined in 15 studies as the shared use of resources between livestock and wildlife, such as feed and water. There were large variations between studies in the defined distances and time windows, with direct contact distances ranging from physical contact (seven studies) to within 120 metres of each other (one study), and indirect definitions ranging from within the same camera image (two studies) to within 50 kilometres of a location (one study). There was less variation in definitions between studies with similar contexts and aims. For example, among M. bovis transmission studies in cattle and badgers, the definition of direct contact ranged from physical contact to within two metres (six studies), and indirect contacts were defined as presence on farmland, sharing of resources and visits to badger latrines by cattle (20 studies). Importantly, no definition of contact was provided in 25 studies (44%) that reported direct contacts, and 34 studies (29%) that reported indirect contacts.

Table 2 Definitions of direct contact from a systematic review of studies of livestock and wildlife.

Parameters are listed in ascending order of distance and time. Definitions that have been used for both direct and indirect contacts are shaded grey. Percentages are rounded to the nearest integer.

‘Direct contact’ definition	Number (%) of publications using this definition	% Cumulative	References	
At least two individuals making physical contact	9 (16)	16	Tolhurst et al. (2009); Brook et al. (2013); Hockings et al. (2012); Campbell et al. (2019); Tolhurst, Ward & Delahay (2011); Vercauteren et al. (2007a); Vercauteren et al. (2007b); Jori et al. (2017); Trabucco et al. (2013)	
Individuals close enough to inhale expired breath	1 (2)	18	Benham & Broom (1989)	
Individuals within one metre of the same location within one second	1 (2)	20	Lavelle et al. (2016)	
Individuals within two metres of each other	5 (9)	29	Drewe et al. (2013); Cowie et al. (2016); Drewe et al. (2012); Garnett, Delahay & Roper (2002); Woodroffe et al. (2016)	
Individuals within five metres of each other	3 (5)	34	Böhm, Hutchings & White (2009); Walter et al. (2012); Hill (2005)	
Individuals within the same camera image	5 (9)	43	Kukielka et al. (2013); Balseiro et al. (2019); Barasona et al. (2017); Cadenas-Fernández et al. (2019); Payne et al. (2016)	
Individuals within 20 metres of each other	1 (2)	45	Richomme, Gauthier & Fromont (2006)	
Individuals within 20 metres of the same location within 15 min	1 (2)	46	Cooper et al. (2010)	
Individuals within same farm building	1 (2)	48	Fleming et al. (2016)	
Individuals within holding (farm) boundary	1 (2)	50	Wu et al. (2012)	
Individuals within 100 metres of each other	2 (4)	54	Wyckoff et al. (2009); Dion, VanSchalkwyk & Lambin (2011)	
Individuals within 120 metres of each other	1 (2)	55	Kukielka et al. (2016)	
Studies that reported the frequency of, types of, or risk factors for, direct contacts without first defining them	25 (45)	100	Ausband et al. (2013); Mizutani, Kadohira & Phiri (2012); Munyeme et al. (2010); Witmer et al. (2010); Molla et al. (2013); Pruvot et al. (2014); Mattiello et al. (2002); Arzamendia & Vilá (2015); Colman et al. (2012); Kolowski & Holekamp (2006); Laporte et al. (2010); Mattiello et al. (1997); Steyaert et al. (2011); Rüttimann, Giacometti & McElligott (2008); Schroeder et al. (2013); Stahl et al. (2002); Anderson et al. (2019); Barasona et al. (2013); Carrasco-Garcia et al. (2016); Carusi, Beade & Bilenca (2017); Howe et al. (2000); Meunier et al. (2017); Ward et al. (2008); Weise et al. (2019); Zarco-González & Monroy-Vilchis (2014)	
Total	56 (100)			

Table 3 Definitions of indirect contact from a systematic review of studies of livestock and wildlife.

Parameters are listed in ascending order of distance and time. Definitions that have been used for both direct and indirect contacts are shaded grey. Percentages are rounded to the nearest integer.

‘Indirect contact’ definition	Number (%) of publications using this definition	% Cumulative	References	
Individuals within the same camera image	2 (2)	2	Payne et al. (2016); Kaczensky et al. (2019)	
Two individuals photographed by the same camera trap within a specific time interval	1 (1)	3	Kukielka et al. (2013)	
Latrine (faecal pits) visits	5 (4)	7	Drewe et al. (2013); Drewe et al. (2012); Scantlebury et al. (2004); Smith et al. (2009); Hutchings & Harris (2009)	
Individuals visiting the same food or water source at the same time	2 (2)	9	Munyeme et al. (2010); Jori et al. (2017)	
Individuals visiting the same food and water sources at unspecified time intervals	13 (11)	20	Lavelle et al. (2016); Katale et al. (2013); Brook et al. (2013); Trabucco et al. (2013); Garnett, Delahay & Roper (2002); Walter et al. (2012); Balseiro et al. (2019); Barasona et al. (2013); Carrasco-Garcia et al. (2016); Brook (2010); O’Mahony (2014); Atwood et al. (2009); Tsukada et al. (2010)	
Individuals in the same space at the same time	2 (2)	22	Maleko et al. (2012); Mullen et al. (2013)	
Individuals in the same space at different times	3 (3)	24	Raizman et al. (2013); Campbell et al. (2019); Barth et al. (2018)	
Individuals in the same space at unspecified time interval	3 (3)	27	Barasona et al. (2014); Robertson et al. (2019); Hill (2005)	
Individuals using the same food or water source within six hours	1 (1)	28	Cowie et al. (2016)	
Individuals within 20 metres of the same location within six hours	1 (1)	28	Cooper et al. (2010)	
Individuals within 30 metres of livestock or feed	1 (1)	29	Ribeiro-Lima et al. (2017)	
Presence in farm buildings at unspecified time interval	5 (4)	34	Witmer et al. (2010); Tolhurst et al. (2009); Tolhurst, Ward & Delahay (2011); Robertson et al. (2017); Woodroffe et al. (2017)	
Individuals within 50 metres of each other	1 (1)	34	Rüttimann, Giacometti & McElligott (2008)	
	
Individuals within 52 metres of the same location within one hour	1 (1)	35	Triguero-Ocaña et al. (2019)	
	
Individuals within 120 metres	1 (1)	36	Brahmbhatt et al. (2012)	
	
Individuals using the same space with seven days	2 (2)	38	Crawford et al. (2019); Cadenas-Fernández et al. (2019)	
Individuals using the same space within 15 days	1 (1)	39	Richomme, Gauthier & Fromont (2006)	
	
Presence on pasture at the same time	5 (4)	43	Clifford et al. (2009); Benham & Broom (1989); Carusi, Beade & Bilenca (2017); Weise et al. (2019); Zarco-González & Monroy-Vilchis (2014)	
Presence on pasture at unspecified time interval	8 (7)	50	Pruvot et al. (2014); Fleming et al. (2016); Guillermo Bueno et al. (2010); Woodroffe et al. (2016); Bromen et al. (2019); Chavez & Gese (2006); Ham et al. (2019); Muhly et al. (2010)	
Presence on pasture at different times	1 (1)	51	Odadi et al. (2017)	
	
At holding boundary and on pasture at unspecified time interval	1 (1)	52	Gehring et al. (2010)	
Presence on farm at unspecified time interval	12 (10)	62	Mullen et al. (2015); Sleeman, Davenport & Fitzgerald (2008); O’brien et al. (2014); O’Mahony (2015); Anderson et al. (2019); Ward et al. (2008); Braz et al. (2019); Judge et al. (2011); Kamler et al. (2019); Van Der Weyde et al. (2017); Viggers & Hearn (2005); Berentsen et al. (2014)	
At holding (farm) boundary	3 (3)	65	Vercauteren et al. (2007a); Vercauteren et al. (2007b); Tolhurst et al. (2008)	
Individuals within 120 metres of the same location at different times	1 (1)	66	Kukielka et al. (2016)	
Individuals within 300 metres of the same location within 15 days	2 (2)	67	Miguel et al. (2013); Miguel et al. (2017)	
Individuals within 500 metres of the same location within six weeks	1 (1)	68	Meunier et al. (2017)	
	
Individuals within 500 metres from holding (farm) boundary	2 (2)	70	Wu et al. (2012); Wyckoff et al. (2009)	
Individuals within 50 kilometres of the same location within three months	1 (1)	71	Beauvais et al. (2019)	
Studies that reported the frequency of, types of, or risk factors for, indirect contacts without first defining them	34 (29)	100	Mizutani, Kadohira & Phiri (2012); Sitters et al. (2009); Molla et al. (2013); Cooper et al. (2008); Oriol-Cotterill et al. (2015); Mattiello et al. (2002); Arzamendia & Vilá (2015); Colman et al. (2012); Kolowski & Holekamp (2006); Laporte et al. (2010); Mattiello et al. (1997); Steyaert et al. (2011); Barasona et al. (2013); Howe et al. (2000); Abade et al. (2018); Acebes, Traba & Malo (2012); Atickem & Loe (2014); Borgnia, Vilá & Cassini (2008); Coe et al. (2001); Cohen et al. (1989); Dohna et al. (2014); Engeman, Betsill & Ray (2011); Jori et al. (2009); Kitts-Morgan et al. (2015); Knust, Wolf & Wells (2011); Kuiters, Bruinderink & Lammertsma (2005); Loft, Menke & Kie (1986); Moa et al. (2006); Pearson et al. (2014); Salter & Hudson (1980); Shrestha (2007); Valls-Fox et al. (2018); Wronski et al. (2015); Anonymous (2013)	
Total	116 (100)			

Regardless of the contact definitions or methods used to observe contacts, direct contacts were detected much less frequently than indirect contacts. For example, one study (Lavelle et al., 2016) found no instances of cattle within two metres of deer, compared to over 40,000 indirect contacts of deer with cattle via shared feed. Overall, the median number of direct contacts between wildlife and livestock was in single figures, whereas the median number of indirect contacts occurred in the order of hundreds or even thousands. The types of contacts reported between livestock and wildlife, and the methods used to observe contacts, are summarised in Table 4. Low study power was acknowledged, but not calculated, by 11 studies (9%), and is likely to be a feature of many more which did not report it. No studies reported adequate power. The low power of studies to observe rare contacts, coupled with the variation in, or lack of, contact definitions, makes it very difficult to compare the effectiveness of the methods used to observe wildlife-livestock contacts.

Table 4 A summary of the types of contact(s) reported between livestock and wildlife, and the method(s) used to observe contacts, from a systematic review of 122 studies.

Livestock	Wildlife	Method(s)a	Type of contact recorded	Examples of the types of contact(s) reportedbetween each livestock and wildlife species	References	
			Direct	Indirect			
Camelid	Antelope	Multiple (d,k,q)	Yes	Yes	Shared space use	Beauvais et al. (2019)	
	Camelid	Direct visualisation	Yes	Yes	Wild camelids grazing with domestic llamas	Arzamendia & Vilá (2015)	
		Multiple (a,d)	No	Yes	Shared forage	Borgnia, Vilá & Cassini (2008)	
Cattle	Antelope	Activity signs	No	Yes	Shared space use	Atickem & Loe (2014)	
		Direct visualisation	No	Yes	Unspecified contact	Wronski et al. (2015)	
		Modelb	No	Yes	No contacts observed	Howe et al. (2000)	
		Multiple (a,k,q)	Yes	Yes	Shared space use	Beauvais et al. (2019)	
		Questioning	Yes	Yes	Shared space use. Shared grazing and water source	Meunier et al. (2017)	
	Badger	Activity signs	No	Yes	Cattle investigating or grazing at badger latrines and setts on pasture	Scantlebury et al. (2004)	
		Camera	Yes	Yes	Badgers and cattle being within two metres of each other. Cattle investigating badger setts and latrines. Badgers visiting farms, feed stores and cattle houses and foraging on cattle pasture. Shared use of water and feed troughs	O’Mahony (2015); Campbell et al. (2019); Payne et al. (2016); O’Mahony (2014); Tsukada et al. (2010); Judge et al. (2011); Anonymous (2013)	
		Direct visualisation	Yes	Yes	Badgers foraging on cattle pasture	Benham & Broom (1989)	
		GPS	No	Yes	Badger visits to cattle farms. Badgers and cattle being present on pasture at the same time, and at different times	Mullen et al. (2015); Mullen et al. (2013); Ham et al. (2019)	
		Model	No	Yes	Cattle grazing at or investigating badger latrines	Smith et al. (2009); Hutchings & Harris (2009)	
		Multiple (a,c,m)	Yes	Yes	Badgers and cattle being within two metres of each other. Badgers visiting feed stores and shared use of feed and water troughs	Garnett, Delahay & Roper (2002)	
		Multiple (a,c,r)	No	Yes	Badgers in and around cattle buildings	Robertson et al. (2017)	
		Multiple (a,q)	No	Yes	Badgers visiting cattle housing, feed stores and feed and water troughs	Robertson et al. (2019)	
		Multiple (a,c)	No	Yes	Badgers visiting farmyards	Sleeman, Davenport & Fitzgerald (2008)	
		Multiple (d,c,r)	No	Yes	Badgers visiting farm boundaries	Tolhurst et al. (2008)	
		Multiple (c,g)	Yes	Yes	Nose to nose contact. Badgers visiting farmyards, farm buildings and feed stores and eating cattle feed	Tolhurst et al. (2009)	
		Multiple (c,q)	Yes	Yes	Nose to nose contact. Badgers visiting, urinating and defecating in farmyards, farm buildings and feed stores and eating cattle feed	Ward et al. (2008)	
		Multiple (c,r)	Yes	Yes	Shared space use	Woodroffe et al. (2016)	
		Multiple (c,l)	Yes	Yes	Shared use of feed troughs	Woodroffe et al. (2017)	
		Proximity logger	Yes	Yes	Badgers and cattle being within one to two metres of each other. Cattle visits to badger latrines	Böhm, Hutchings & White (2009); Drewe et al. (2013); Drewe et al. (2012)	
	Big cat	Camera	No	Yes	No contacts observed	Abade et al. (2018)	
		GPS	No	Yes	Lion presence on cattle pasture. Cheetah visits to cattle farms	Weise et al. (2019); Van Der Weyde et al. (2017)	
		Multiple (a,c)	Yes	Yes	Predation events and wild felid presence oncattle pasture	Zarco-González & Monroy-Vilchis (2014)	
	Buffalo	GPS	No	Yes	Shared space and water sources	Miguel et al. (2013); Miguel et al. (2017); Valls-Fox et al. (2018)	
		Model	Yes	No	Cattle and buffalo being within 100 metres of each other	Dion, VanSchalkwyk & Lambin (2011)	
		Literature review	No	Yes	Young buffalo joining cattle herd and ’contact’ (unspecified) between cattle and buffalo	Jori et al. (2009)	
		Questioning	Yes	Yes	Shared grazing and water source	Meunier et al. (2017)	
	Camelid	Activity signs	No	Yes	No contacts observed	Acebes, Traba & Malo (2012)	
		Direct visualisation	Yes	No	No contacts observed	Arzamendia & Vilá (2015); Schroeder et al. (2013)	
		Multiple (a,d)	No	Yes	Shared forage	Borgnia, Vilá & Cassini (2008)	
	Canine	Camera	Yes	Yes	Cattle and foxes being within two metres of each other. Foxes visiting farm buildings, foraging and hunting on farmland and defecating on stored feed	Tolhurst, Ward & Delahay (2011)	
		GPS	Yes	Yes	Wolf visits to cattle pasture	Laporte et al. (2010); Steyaert et al. (2011); Muhly et al. (2010)	
		Multiple (a,d)	No	Yes	Wolf and coyote presence on cattle pasture	Gehring et al. (2010)	
		Radio-telemetry	No	Yes	Wolf visits to cattle pasture. Jackal visits to cattle farms	Chavez & Gese (2006); Kamler et al. (2019)	
	Deer	Activity signs	No	Yes	Deer presence on pasture previously grazed by cattle	Kuiters, Bruinderink & Lammertsma (2005)	
		Camera	Yes	Yes	Shared use of feed and water troughs	Kukielka et al. (2013); Payne et al. (2016); Barasona et al. (2013); Carrasco-Garcia et al. (2016); Tsukada et al. (2010)	
		Direct visualisation	Yes	Yes	Aggression between deer and cattle, and deer and cattle being within five metres of each other. Deer visits to cattle feed stores and deer presence on pasture at the same time and at different times to cattle. Deer licking cattle urine	Mattiello et al. (2002); Hill (2005); Mattiello et al. (1997); Carusi, Beade & Bilenca (2017)	
		GPS	No	Yes	Deer visits to cattle pastures and feeding areas	Cooper et al. (2008); Ribeiro-Lima et al. (2017); Berentsen et al. (2014)	
		Literature review	Yes	Yes	No contacts observed	Walter et al. (2012)	
		Multiple (a,c)	Yes	Yes	Cattle and deer at water sources at the same time	Barasona et al. (2017)	
		Multiple (d,c)	Yes	Yes	Unspecified contact	Brook et al. (2013)	
		Multiple (c,p)	Yes	Yes	Cattle and deer within 1.5 metres of each other. Shared use of water and food points	Cowie et al. (2016)	
		Multiple (g,l)	No	Yes	Deer presence on cattle pasture	Gehring et al. (2010)	
	
		Multiple (g,q)	Yes	Yes	Unspecified direct contact. Deer visits to cattle feed stores	Pruvot et al. (2014); Kitts-Morgan et al. (2015)	
		Proximity logger	Yes	Yes	Deer visits to stored feed	Lavelle et al. (2016)	
		Questioning	No	Yes	Deer presence on cattle farms, and visiting and damaging feed stores	Brook (2010); Knust, Wolf & Wells (2011)	
		Radio-telemetry	No	Yes	Deer visits to cattle pasture and shared salt licks	Coe et al. (2001); Cohen et al. (1989); Dohna et al. (2014); Loft, Menke & Kie (1986)	
	Elephant	GPS	No	Yes	Elephant home range overlapping with cattle grazing. Elephants using same water source at the same time and at different times to cattle	Raizman et al. (2013); Valls-Fox et al. (2018)	
	Hyena	Multiple (d,r)	Yes	Yes	Predation events	Kolowski & Holekamp (2006)	
	Kangaroo	Radio-telemetry	No	Yes	Kangaroo presence on cattle farms	Viggers & Hearn (2005)	
	Not specified	Camera	Yes	Yes	Raccoons licking salt lick less than thirty centimetres away from cattle, and sharing water sources. Savannah wildlife grazing at the same and at different times to cattle	Crawford et al. (2019); Witmer et al. (2010); Odadi et al. (2017)	
		Direct visualisation	Yes	Yes	Cattle and savanna wildlife sharing water sources at the same and at different times	Mizutani, Kadohira & Phiri (2012)	
		Questioning	Yes	Yes	Wildlife and cattle sharing water sources and grazing at the same and at different times	Munyeme et al. (2010); Katale et al. (2013); Brahmbhatt et al. (2012); Molla et al. (2013); Maleko et al. (2012)	
		Radio-telemetry	No	Yes	No contacts observed	Sitters et al. (2009)	
	Raccoon	Multiple (c,l,r)	No	Yes	Shared space use. Shared food and water sources	Atwood et al. (2009)	
	Sheep/Goat	Direct visualisation	Yes	Yes	Chamois and ibex in close proximity to cattle. Shared use of cattle pasture	Richomme, Gauthier & Fromont (2006)	
		Multiple (g,m)	No	Yes	No contacts observed	Clifford et al. (2009)	
	Wild horse	GPS	No	Yes	Spatial overlap of zebra home ranges with cattle grazing areas. Shared use of water source	Raizman et al. (2013)	
		Multiple (a,d)	No	Yes	Feral horses grazing in close proximity to cattle, and using pasture prior to cattle	Salter & Hudson (1980)	
	Wild pig	Activity signs	No	Yes	Wild boar presence on pasture previously grazed by cattle	Guillermo Bueno et al. (2010); Kuiters, Bruinderink & Lammertsma (2005)	
		Camera	Yes	Yes	Wild boar and cattle sharing water sources and feed troughs at the same time and at different times	Kukielka et al. (2013); Balseiro et al. (2019); Payne et al. (2016); Barasona et al. (2013); Carrasco-Garcia et al. (2016); Tsukada et al. (2010)	
		GPS	No	Yes	Shared space and water sources	Barasona et al. (2014); Triguero-Ocaña et al. (2019)	
		Multiple (c,g)	Yes	Yes	Wild boar and cattle sharing water source at the same time	Barasona et al. (2017)	
		Multiple (c,p)	Yes	Yes	Feral pigs and cattle being within 20 metres of the same location at different times	Cooper et al. (2010)	
		Multiple (g,l)	Yes	Yes	Wild boar and cattle being within 1.5 metres of each other. Shared use of food and water points	Cowie et al. (2016))	
		Questioning	Yes	Yes	Shared water sources	Anderson et al. (2019); Meunier et al. (2017)	
Farmed deer	Big cat	Radio-telemetry	No	Yes	No contacts observed	Moa et al. (2006)	
	Deer	Camera	Yes	Yes	Sparring and nose to nose contact, and presence of wild deer at fence-line of farmed deer	Vercauteren et al. (2007a); Vercauteren et al. (2007b)	
Goat	Antelope	Multiple (d,k,q)	Yes	Yes	Shared space use	Beauvais et al. (2019)	
	Big cat	Camera	No	Yes	No contacts observed	Abade et al. (2018)	
		Multiple (a,c)	Yes	Yes	Predation events and wild felid presence on goat pasture	Zarco-González & Monroy-Vilchis (2014)	
	Camelid	Direct visualisation	Yes	Yes	Shared forage sources at different times	Arzamendia & Vilá (2015); Schroeder et al. (2013)	
		Multiple (a,d)	No	Yes	Shared forage	Borgnia, Vilá & Cassini (2008)	
	Canine	Radio-telemetry	No	Yes	Jackal visits to goat farms	Kamler et al. (2019)	
	Chimpanzee	Direct visualisation	Yes	No	No contacts observed	Hockings et al. (2012)	
	Deer	Camera	Yes	Yes	No contacts observed	Carrasco-Garcia et al. (2016)	
	Hyena	Multiple (d,r)	Yes	Yes	Predation events	Kolowski & Holekamp (2006)	
	Not specified	Camera	No	Yes	Presence on pasture of predators not associated with livestock predation	Bromen et al. (2019)	
	Wild pig	Camera	Yes	Yes	No contacts observed	Carrasco-Garcia et al. (2016)	
		Questioning	Yes	Yes	Predation and presence on farm	Anderson et al. (2019)	
Not specified	Big cat	GPS	No	Yes	No contacts observed	Oriol-Cotterill et al. (2015)	
	Sheep/Goat	Direct visualisation	No	Yes	Shared space use and forage	Shrestha (2007)	
	Wild horse	Multiple (c,g)	Yes	Yes	Livestock within photographing distance of khulan horses	Kaczensky et al. (2019)	
Pig	Canine	Camera	Yes	Yes	Foxes approaching and entering farrowing huts and taking piglets. Fox presence in pig paddocks	Fleming et al. (2016)	
	Deer	Camera	Yes	Yes	Shared water sources	Kukielka et al. (2013); Carrasco-Garcia et al. (2016)	
		Multiple (g,l)	Yes	Yes	Deer and pigs within 1.5 metres of each other. Shared use of food and water	Cowie et al. (2016)	
	Wild pig	Camera	Yes	Yes	Shared food and water sources. Wild boar visiting acorn fields used by domestic pigs	Kukielka et al. (2013); Carrasco-Garcia et al. (2016)	
		GPS	No	Yes	No contacts observed	Pearson et al. (2014)	
		Multiple (a,c,g)	No	Yes	Wild boar home range overlap with domestic pigs and shared space use	Barth et al. (2018)	
		Multiple (a,c,q)	No	Yes	No contacts observed	Braz et al. (2019)	
		Multiple (c,m)	Yes	Yes	Pigs and wild boar present in the same camera trap image. Shared use of water	Cadenas-Fernández et al. (2019)	
		Multiple (c,q)	Yes	Yes	Wild boar and pigs within 1.5 metres of each other. Shared use of food and water	Cowie et al. (2016)	
		Multiple (g,l)	No	Yes	Feral swine presence around pig farms	Engeman, Betsill & Ray (2011)	
		Multiple (m,q)	Yes	Yes	Evidence of mating (cross-bred piglets). Wild boar within two metres of pig enclosure	Wu et al. (2012)	
		Multiple (p,r)	Yes	Yes	Feral and domestic swine in contact through fences. Feral pigs within 500 metres of pig farm	Wyckoff et al. (2009)	
		Questioning	Yes	Yes	Wild and domestic pigs fighting and mating. Shared use of water, food and space at different times	Jori et al. (2017); Trabucco et al. (2013); Kukielka et al. (2016); Anderson et al. (2019)	
Sheep	Antelope	Multiple (d,k,q)	Yes	Yes	Shared space use	Beauvais et al. (2019)	
	Badger	GPS	No	Yes	Badger visits to sheep farms	Mullen et al. (2015)	
	Big cat	Radio-telemetry	Yes	Yes	Predation	Stahl et al. (2002); Moa et al. (2006)	
	Camelid	Direct visualisation	Yes	Yes	Shared forage sources at different times	Arzamendia & Vilá (2015); Schroeder et al. (2013)	
		Multiple (a,d)	No	Yes	Shared forage	Borgnia, Vilá & Cassini (2008)	
	Canine	GPS	Yes	No	No contacts observed	Ausband et al. (2013)	
		Radio-telemetry	No	Yes	Jackal visits to sheep farms	Kamler et al. (2019)	
	Chimpanzee	Direct visualisation	Yes	No	No contacts observed	Hockings et al. (2012)	
	Deer	Camera	Yes	Yes	No contacts observed	Carrasco-Garcia et al. (2016)	
		Direct visualisation	Yes	Yes	Deer and sheep within five metres of each other	Colman et al. (2012)	
	Hyena	Multiple (d,r)	Yes	Yes	Predation events	Kolowski & Holekamp (2006)	
	Kangaroo	Radio-telemetry	No	Yes	Kangaroo visits to sheep farms	Viggers & Hearn (2005)	
	Not specified	Camera	No	Yes	Presence on pasture of predators not associated with livestock predation	Bromen et al. (2019)	
	Sheep/Goat	Direct visualisation	Yes	Yes	Chamois and ibex in close proximity to domestic sheep and sharing pasture	Richomme, Gauthier & Fromont (2006); Rüttimann, Giacometti & McElligott (2008)	
		Radio-telemetry	No	Yes	Unspecified contact	O’brien et al. (2014)	
	Wild pig	Activity signs	No	Yes	Wild boar foraging on sheep pasture	Guillermo Bueno et al. (2010)	
		Camera	Yes	Yes	No contacts observed	Carrasco-Garcia et al. (2016)	
		Questioning	Yes	Yes	Predation and presence on sheep farms	Anderson et al. (2019)	
Notes.

a Some studies used multiple methods combining variations of activity signs (a), cameras (c), Direct visualisation (d), GPS (g), literature review and expert knowledge elicitation (k), models (m), pathogen monitoring (p), proximity loggers (l), questioning (q) and radio-telemetry (r).

b Where modelling alone is reported, empirical data was used that was not specifically wildlife-livestock contact data. For example, using data on cattle grazing habits to model the frequency of contact with badger faeces on pasture.

Proposed unified framework to define direct and indirect contacts

Space (area or distance between animals) and time were crucial components of the varied definitions of direct and indirect contact in this review. In an effort to unify these parameters, a novel generic framework to categorise wildlife-livestock contacts is proposed in Fig. 4, based on the locations of individuals in space and time. Using this framework, we propose that the contact type (direct or indirect) is defined using the two parameters SC and TC. Multiple critical thresholds can be used within the framework to differentiate between definitions of direct contact (SC1 and TC1) and indirect contact (SC2 and TC2). For a direct contact to occur, two individuals are within the same pre-specified critical space (distance or area: SC1) within a pre-specified critical time window (TC1). Similarly, for an indirect contact to occur, animals are within another pre-specified critical space (SC2) within another pre-specified critical time window (TC2). The reader is directed to Fig. 4 for examples from the literature of possible combinations of SC and TC. TC2 may be the same as TC1 (if SC2 is larger than SC1: compare example A with example B in Fig. 4) or TC2 may be different from TC1(in which case TC2 will usually, but not always, be larger than TC1: compare example A with examples C, D, E and F in Fig. 4). Similarly, SC2 may be the same as SC1 (if TC2 is larger than TC1: compare example A with examples C and E in Fig. 4) or SC2 may be different from SC1 (in which case SC2will usually, but not always, be larger than SC1: compare example A with examples B, D and F in Fig. 4). Same, near and different are used here to illustrate spatial and temporal differences between examples. These terms are relative and will vary along with SC and TC depending on the system being studied, the objectives of the study and other factors such as host behaviour and the biology of the pathogen, in the case of disease studies; therefore, values for TC1, TC2, SC1 and SC2 should be decided in advance of a study being conducted, and they should be clearly reported when data are presented.

Figure 4 A proposed generic framework for describing and categorising contacts between livestock and wildlife.

Examples from studies of contacts between badgers and cattle are provided to demonstrate the use of the framework. SC1 represents ‘critical space 1’ , the maximum amount of space (distance or area) within which direct contact may occur; and TC1 represents ‘critical time 1’ , the maximum duration of time within which direct contact may occur. Similarly, SC2 represents ‘critical space 2’ , the maximum amount of space (distance or area) within which indirect contact may occur; and TC2 represents ‘critical time 2’ , the maximum duration of time within which indirect contact may occur. Same, near and different are used here to illustrate spatial and temporal differences between examples (see Tables 2–4 for values and ranges for these parameters from published studies). Note that the lighter blue shading does not extend all the way to the right of the diagram because there is an upper limit to the value of time which TC2 can take: beyond this value, animals in the same (or nearby) space will not be in contact. Ref a = Tolhurst et al. (2009), ref b = Benham & Broom (1989), ref c = Drewe et al. (2013), ref d = Woodroffe et al. (2016), ref e = O’Mahony (2015), ref f = Mullen et al. (2015).

Although the exact values of the critical distance between animals and the critical time window over which this happens will depend on the system being studied as well as the specific objectives of each study, the adoption of this generic framework to define direct and indirect contacts will help identify studies with similar definitions where results are more easily comparable.

Discussion

The need for a generic unified framework

This review has found that definitions of contact are wide-ranging and highly dependent on the context of the study. Definitions can vary depending on the species and demographics of the wildlife and livestock involved, the methods used to detect contacts and the system being studied such as the environmental conditions and pathogen characteristics in studies where contacts are representative of disease transmission. Definitions of direct contact were extremely diverse, ranging from direct physical contact to animals being merely within a hundred metres of each other. Indirect contact ranged from animals sharing resources, being within five kilometres of each other or overlapping in home ranges, and the time window that these events occurred in varied from hours to weeks.

The aim of this generic unified framework is to promote consistent reporting of definitions of contacts enabling comparisons to be made between the approaches of wildlife-livestock contact studies, regardless of the species or pathogen studied or the context of the study. This is needed because our systematic review found that while wildlife-livestock contact data was collected in terms of space and time, some studies omitted space or time in their definitions, or there was a complete lack of a definition. Conflicting and overlapping definitions of direct and indirect contact were also identified. Making any sort of meaningful comparison between such studies is challenging. For example it is difficult to assess what, if any, implications there are for deer-cattle disease transmission from a behavioural study showing deer avoid cattle despite similar habitat preferences (Mattiello et al., 2002), without knowing what types of contact (e.g., direct or indirect; what specific types) were likely to be meaningful. It is even difficult to compare studies within the same system, for example establishing the relevance of cattle-badger contacts for bovine tuberculosis transmission when some studies define a contact as ‘presence on farm’ (Mullen et al., 2015; Sleeman, Davenport & Fitzgerald, 2008) and others define it as ‘presence in buildings’, and neither study defines the time window. Use of the generic unified framework would enable consistent reporting of definitions between studies and is an important step if the results of wildlife-livestock contact studies are to be comparable.

Applications of a generic unified framework

Models that incorporate empirical rather than theoretical information on the frequency and duration of contacts important for disease transmission are more likely to be useful for disease mitigation (Craft, 2015). The use of a standardised definition framework in future studies of livestock-wildlife contacts would enable consistency in datasets and enable the retrospective selection of contact data relevant to a particular model, which could then be incorporated in a similar way to the data used in recent bovine tuberculosis transmission models (Wilber et al., 2019; Silk et al., 2018). The generic unified framework proposed in this current paper could also be useful in designing livestock-wildlife contact studies, since defining the type of contact to be detected—in addition to practical considerations, such as an area’s signal strength affecting the viability of GPS device use—helps with the choice of detection method. The framework is also flexible and applicable to different contexts, species and diseases since it allows for the variation in definitions seen in this review, and it is hoped it will broaden the range of future livestock-wildlife contact studies.

To resolve human-wildlife conflicts usually requires robust livestock-wildlife contact studies. At least 120 studies that only used predation events to infer livestock-wildlife contacts were excluded from the review, yet predators –particularly wolves –were the second most commonly studied group of wild mammals. Given that predation studies appear to form a large proportion of wildlife-livestock contact studies, this is an area where adoption of the generic framework could help design meaningful contact studies to evaluate preventive measures without relying solely on predation events.

Further development of the generic unified framework

The generic unified framework does not provide an overall consensus on definitions of direct and indirect contact, but does provides a structure with which to start this process. While using the generic unified framework provides a standardised definition of contact in time and space, identifying the types of contact that are relevant to the study, and thus the values of SC and TC, will vary depending on the objectives and context of each study. While a universally accepted set of definitions for contacts is difficult to devise, we hope that by defining Sc and Tc here we will encourage the start of the debate around (and between) studies of similar contexts, and perhaps then acceptable ranges for these values will emerge. Developing a framework for deriving SC and TC, based upon the species studied, environment, pathogen and methodology is beyond the scope of this review, and would be a necessary next step so that wildlife-livestock contact rates could be comparable between studies of similar contexts. For example, for disease studies, it would be advisable that SC and TC were based on values below which transmission is likely to occur, such as aerosol dispersion distance and environmental survivability. For any system, there may be a range of appropriate values for SC and TC.

The generic unified framework presented in this paper is a step towards being able to compare observation methods and contact data in order to standardise and evaluate different monitoring methods. This is important as our systematic review revealed that the methods used to observe livestock-wildlife contacts to date have often had low detection rates and therefore been of low power due to the difficulty of monitoring cryptic wildlife species, and the rarity of some types of wildlife-livestock contacts, particularly direct contacts. Further considerations for the comparison of observation methods are the representativeness of individuals monitored, especially with methodologies that require the marking of individuals such as GPS and proximity loggers, and a standardised system for relativizing the number of contacts with regards to the total observation effort. For example, two studies will not be comparable if study A only uses 3 camera traps and study B uses 100 camera traps, or if study C collects GPS locations every hour when study D collects only one GPS fix per day. Reporting representativeness of individuals and relativizing contact rates in terms of total population will go some way to establishing the power of wildlife-contact studies. Furthermore, it may be useful for studies to indicate the detection limits of the methodology used, in terms of space and time.

Scope of existing wildlife-livestock contact studies

This review has identified the narrow scope and limited geographic range of livestock-wildlife contact studies, which means the data summarised in this review should not be considered representative of all wildlife-livestock contacts worldwide. The majority of studies focussed on cattle-wildlife contacts and diseases of cattle. Bovine tuberculosis (infection with M. bovis) featured prominently, indicative of the economic and potentially zoonotic importance of this disease to the USA and UK, where the most livestock-wildlife contact studies were conducted (De la Rua-Domenech, 2006; O’Brien et al., 2011). Foot-and-mouth-disease was the most studied viral pathogen and this is most likely explained by its broad geographical spread and high economic impact (Knight-Jones & Rushton, 2013). This demonstrates the human-centric view of the wildlife-livestock interface, with most focus on the impacts on humans and domestic animals, and very little (if any) focus on the value of wildlife (Chardonnet et al., 2002). There were, however, some livestock-wildlife contact studies of high impact conservation importance such as infection with Mannheimia spp. in bighorn sheep (Ovis canadensis) and Pasteurella spp. in saiga antelope (Clifford et al., 2009; O’brien et al., 2014; Beauvais et al., 2019). If we are to collect more (and better) wildlife-livestock contact data that include a broader range of species and contexts, careful consideration must be used when selecting the most effective and practical observational method for monitoring cryptic wildlife species.

This review highlights that observing contacts between multiple species is possible and can yield high quality information. Increasing the efficiency of monitoring methods would justify their use for more applications. Health surveillance systems at livestock-wildlife interfaces have been suggested as a method to detect and control emerging diseases along with preventing contact between wildlife and livestock (Gortazar et al., 2015). Preventing high-risk contacts may be more cost-effective than surveillance, but the effectiveness of prevention strategies will need to be evaluated by monitoring contacts, or lack thereof. More efficient monitoring will also allow for quantitative risk assessments of wildlife-livestock contacts which are presently difficult to conduct due to a limited understanding of potential contacts leading to pathogen transmission (Miller, Farnsworth & Malmberg, 2013). Some observation methods such as camera traps are likely to have the ability to identify new potential transmission routes between livestock and wildlife (e.g., the use of cattle salt licks by raccoons (Witmer et al., 2010)), and may identify livestock-wildlife contacts previously not considered (e.g., observing farm visits by foxes during a study focussing on badgers (O’Mahony, 2015)). Identifying wildlife species that may be the origin of rapidly emerging human diseases is a priority to prevent future pandemics (Morse et al., 2012). In situations where human infections are mediated by livestock, rapid implementation of observational methods to detect contacts between wildlife and livestock could more quickly identify wildlife hosts and risky behaviours. In order to determine the efficiency and efficacy of different observational methods, the methods used and data collected by them must be comparable, hence the need for a unified framework.

Limitations of this review

Our study has some limitations which we summarise here. At present, our generic unified framework does not explicitly account for disease transmission via vectors or fomites, although the latter will to some extent be captured within our definition of indirect contact. In order that observation methods were likely to be comparable between species, we focussed on terrestrial mammals so did not address diseases primarily hosted by birds or bats such as avian influenza, Nipah virus and Hendra virus. Small terrestrial mammals (<5 kg) were also not included for this reason, and because a disproportionate number of rodent studies focus on their roles as laboratory animals or farm pests, and not on contacts with livestock. While the generic unified framework may be applicable to these types of wildlife, it is unclear which observational methods seen in this review would be most effective or efficient, and the conclusions drawn from this review may not be reflective of systems that involve other taxa.

Conclusion

As human populations continue to expand and agriculture encroaches further on wildlife habitats, disease spill-over (in both directions) between wildlife, livestock and humans is becoming more frequent (Wiethoelter et al., 2015). As a result, the study of contacts between livestock and wildlife is receiving ever increasing attention. This systematic review of the observational methods used to study contacts, and the subsequent proposal of a generic unified framework for defining contacts, are two steps towards ensuring that data are collected and reported in a standardised way at a time of increasingly urgent need.

Supplemental Information

Supplemental Information 1 Rationale for conducting the systematic review, and the contribution to knowledge

Click here for additional data file.

Supplemental Information 2 PRISMA checklist

Click here for additional data file.

Supplemental Information 3 Details of publications included in the systematic review

Click here for additional data file.

Supplemental Information 4 Data extracted from the publications included in the systematic review

This dataset contains raw and coded data used in the analyses. It includes descriptions and aims of 122 publications, the livestock and wildlife species studied and the observation methods used, the types and frequencies of direct and indirect contacts recorded and the definitions used for each, and the emergent themes.

Click here for additional data file.

Supplemental Information 5 Themes of the review

Livestock-wildlife studies (n = 122) grouped by themes that emerged during data extraction. Where studies had more than one theme, each theme was identified as either primary (main) or secondary (supportive) based on the aims of the study; hence the total number of primary themes in this figure exceeds the number of studies.

Click here for additional data file.

Supplemental Information 6 Wildlife and livestock species represented in the review

Many publications monitored multiple species of wildlife and livestock and therefore numbers of studies may exceed 100% for some groups.

Click here for additional data file.

Supplemental Information 7 Livestock-wildlife contacts observed

Data from 122 papers included in the systematic review. The size and shade of circles indicate the number of studies in each category. Many publications used more than one method to monitor contacts, and therefore the numbers of studies exceed 100% for some groups.

Click here for additional data file.

Supplemental Information 8 Methods used to monitor different types of contact

Methods used to observe wildlife grouped by methods used alone and in combination with other methods, and grouped by whether direct or indirect contact, or both, was monitored.

Click here for additional data file.

Supplemental Information 9 Search strings used to identify publications that investigated contact between any wild and domestic mammal in Pubmed, Scopus and CAB Abstracts databases

Click here for additional data file.

Supplemental Information 10 R packages used in the systematic review

Click here for additional data file.

Supplemental Information 11 Location of the 122 wildlife-livestock contact studies included in the systematic review, stratified by continent

Click here for additional data file.

Supplemental Information 12 Bacterial diseases studied in the context of wildlife-livestock contact studies

Click here for additional data file.

Supplemental Information 13 Viral diseases studied in the context of wildlife-livestock contact studies

Click here for additional data file.

Reviewed and approved for submission for publication by the Royal Veterinary College: RVC manuscript number PPS_02151.

Additional Information and Declarations

Competing Interests

Author Contributions

Data Availability

The authors declare there are no competing interests.

Sonny A. Bacigalupo conceived and designed the experiments, performed the experiments, analyzed the data, prepared figures and/or tables, authored or reviewed drafts of the paper, and approved the final draft.

Linda K. Dixon, Simon Gubbins and Adam J. Kucharski conceived and designed the experiments, authored or reviewed drafts of the paper, and approved the final draft.

Julian A. Drewe conceived and designed the experiments, prepared figures and/or tables, authored or reviewed drafts of the paper, and approved the final draft.

The following information was supplied regarding data availability:

Details of the publications included in the review and the raw data extracted are available as Supplemental Files.

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
