# Peer review of "Towards a unified generic framework to define and observe contacts between livestock and wildlife: a systematic review"

_PeerJ, doi:10.7717/peerj.10221_

## Round 0.1 · original submission · Major Revisions

While expert reviewers have commented on the large effort undertaken to complete this systematic review they have identified several major issues that need to be addressed before further consideration. Please provide a detailed comment by comment response identifying line by line changes addressing the detailed comments provided by the reviewers.

Reviewer 1 ·

Basic reporting

The MS meet standards

Experimental design

The approach, literature search, filtering and analysis meet the standards

Validity of the findings

A relevant stop forward on standardizing contact data art the wildlife-livestock interface

Additional comments

The aim of this study was to systematically review the reasons for, and observational methods used in, studies investigating livestock-wildlife contacts, and to propose a generalized framework for defining contacts between livestock and wildlife. Definitively, as the authors claim, a standardized generic template for defining livestock-wildlife contacts is needed. And this MS is an important step forward.
The MS is excellent as a comprehensive review on the topic.The authors show in the results section the novel generic framework to categorise wildlife-livestock contacts (Fig. 4), based on the locations of individuals in space and over time. It shroud previously be introduced also in Material and Methods (the general approach)
L123: legend for Figure S1 (two categories by column??) explain).

L129. Remark that research addressing resource use by wildlife and livestock, but not directly the contacts, is included.

L147. Define “study power”: ability to detect rare contacts??

L148. Authors mention: “Conclusions were robust and directly derived from the results in 109 (89%) publications. Is that subjective”. Provide indication how you assessed this (they all are peer-reviewed papers), did you question conclusions in 11% remaining?

L160: do you include wild boar as wild pig? Specify since it is used over the MS (e.g. fig. 3).

Further analyses (i..e according to system, or geography) of paper scores can provide interesting results.

L202. What about locations first visited by livestock?

L229. The proposed unified framework to define direct and indirect contacts must be introduced (the approach) in Mat & Met), and briefly commented in the summary.

L236: define SC (critical distance between animals, depending on the system being studied as well as the specific objectives of each study, I only found the definition in legend of figure 34) and Tc (critical time over which critical distance happens between 2 individuals, depending on the system being studied as well as the specific objectives of each study) (maybe in Mat & Met).

Discussion.
Proposed unified framework: I agree the approach of this generic framework to define direct and indirect contacts will help ensure results between studies are more easily comparable. However, the main problem: the definition of Sc1, Sc2, Tc2, and Tc2 according to the system and specific conditions. This, itself, needs a common framework (guidance or rules) which is not addressed here and vaguely commented. Then, studies can be comparable. For discussion and future research: I encourage authors to discuss if it can be developed a generic framework to define Sc1,2, Tc1,2 as a function of objective characteristics of the pathogen- host-environmental systems and objectives. This would be also very useful for researchers to approach their study systems to generate comparable values and use the appropriate methodology (and not to make the definition on contact dependent on methods or other practical issues). Provide any clue for what is near or different in Figure 4 for a specific system (this distinction, as it is presented, is vague). Most discussion could be part of the introduction. I encourage authors to expand discussion on the abovementioned, on the limitations of this framework, which need more detail (e.g. definition on near and different) and a parallel framework for definition of Sc1,2, Tc1,2 (in a separate research)

L251: Meaningful comparison only to a certain extend. See my previous comment.

L329: Agree! This is a necessary step forward.

Reviewer 2 ·

Basic reporting

No comment

Experimental design

No comment

Validity of the findings

No comment

Additional comments

The present manuscript performs a systematic review of papers studying wildlife-livestock contacts, and subsequently propose a generalised framework for defining direct and indirect contacts between domestic and wild mammals (Figure 4). The manuscript addresses a highly relevant topic, especially in the current situation of sanitary uncertainty, in which we must increase efforts in sanitary surveillance and in the implementation of control measures of zoonotic diseases. Besides, the review includes the most important published studies of the topic.
However, I have some concerns about the methodology, discussion and the applicability of the proposed framework:

Methodology:
1. Line 88-89: As stated at the end of the paragraph, a contact could be an association between animals and farms, so, this must be included in the formal definition of the contact.
2. Line 96-97 and (Table S1). Just curiosity: regarding the terms included in the searching process, why did you specify so much wild species (lion, antelopes, buffalo, wolf……….) when the most important concepts were also included (i.e. carnivore, herbivore, wildlife….). By being so specific, many species will be forgotten (as for example: giraffe).
3. Line 126: what is the point of defining the quality of the papers? I suggest being more precise on what you mean with quality, since this is a very broad term, and (probably) all of the papers considered in the revision have been published after a peer review, so the quality of all of them is implied.
4. The development of the “generic unified framework” is my major concern. I agree that this is a major weakness in the study of contacts, since many studies are not comparable between them, so the proposal of a unified framework appears like a great solution. However, your proposed framework does not really contribute to the consensus of a generic definition of direct/indirect contacts. This definition has to go beyond a spatio-temporal agreement of what it will be considered as a contact, since this definition will depend on the pathogen (in the case of disease studies), the species, the aim of the study, the methodology employed… For example, an indirect contact (see Figure 4) could be “near in time”; according to this, the temporal window could be 1 hour or 1 day, and this not make comparable the studies using such different windows.

You should also include the problem of the representativeness of the individuals, especially when the contacts have been recorded with a methodology that requires the marking of individuals (GPS, proximity loggers, VHF…). This representativeness will also modify how comparable are two studies.

It is also very important to relativize the number of contacts (both direct and indirect) with regards to the total population. Two studies won’t be comparable if study A only uses 3 camera traps and study B uses 100 camera traps.

I suggest to improve the “proposed unified framework” by including all these considerations, for example, proposing a definition of contact regarding the topic of the study (disease, behaviour…), or regarding the pathogen studied, or the species included….


Results:
1. Line 224: what do you mean with “power”? Explain (maybe in the methodology section?)
Discussion:
2. Line 253: however, you don’t give a real range either
3. Line 271 – 273: this is not precise, since sometimes, the species involved or the study area will condition the methodology employed. For example, we can’t use GPS devices if our study area doesn’t have a proper signal; or we cannot observe directly the contacts if the visibility of our study area is not good. So, the choice of the methodology will depend on many factors, and not only on the spatio-temporal window employed to define contact.
4 Line 273: be careful with the term “observation method” because it could be confused with the direct observation of the contacts (as for example in the case of Richomme et al. 2006)
5. Line 332-333: the direct observation of the contact (which is a methodology included in the review) could be applied in this case


Figures:
1. Figure S1: what does represent the numbers in the bars? The sum of these numbers is higher than 122 (which is the total number of papers included in the review). Besides, what does represent the two themes or categories in this figure (primary and secondary theme)?
2. Figure 4: the title of Figure 4 is too long, especially if we consider that almost all this information appears in the Results section


Minor comments:
1. Line 10: maybe not so surprising, since (as you estate below throughout the text) the definition of contact will depend on many factor: the pathogen under study [transmission route, survival capability…], the species involved, the methodology applied for the recording of contacts….

·

Basic reporting

1. The introduction and background were clear, provided a good foundation for the review, and generally well-referenced. There were several points where clarification was needed (see comments in PDF) in the abstract and results:
a. Lines 13 – 14: 30,080 unique publications were eligible for screening, based on Figure 1. It's more accurate to report the 30,080 rather than 43,000 since many of these were duplicates.
b. Lines 14 – 16 (abstract) vs. 160 – 161 (results): it’s unclear what the most studied wildlife species were (and in what order): deer, wild pigs, or badgers?

2. There are several errors, omissions, typos that need to be fixed in the tables. Please see notes in the PDF. Particular attention should be paid to Table 4, which currently has several errors ranging from issues of formatting to statistical reporting.
a. Your reporting of descriptive statistics needs attention. I'm unsure how you arrived at your summary statistics (mean, median, range). For example, are you reporting the mean number of direct contacts within a single study or are you calculating the mean number of observations within multiple studies? For example, the mean, median, and range are often reported to be the exact same numbers. This is odd and suggests that there is only a single value, in which case there is no need for summary statistics. Also, the range, by definition, has two numbers: a lower limit and an upper limit. Just reporting one number is confusing and incorrect (e.g. PRISMA Checklist #13 is “State the principal summary measures (e.g., risk ratio, difference in means)” and all you say is “Descriptive”).
b. Can you clarify what you mean by “Number of studies” and “References” as column headings (and what the difference is?). For example, on row 3 – 4, I'm confused why the livestock camelid + wildlife camelid rows say that there is only one study, but two different references? Same question applies to numerous rows in the table.
c. A table should be used to summarize and this table is too large and doesn’t condense information effectively. One possible solution is to just reserve this table for actual summarizing of studies involving > 2 studies? Can you further edit or summarize the last column (type of contact). Or perhaps just include publications that actually provide information on the frequency of contacts (i.e. no rows with “-" in them?)
d. I am unsure why “models” are included as methods used to observe contact. Models are either based on empirical data (from observation methods like the ones you mentioned) or theoretical information. Can you explain why you consider “models” an observational method?

3. I think it’s important to mention what “themes” emerged in the actual text of the paper (lines 122 – 123) and then refer the reader to the Supplemental figure to see the breakdown. For example, I didn’t know what themes you were referring to until the results section, but it should be mentioned earlier.

4. Please use caution when reporting percentages and mention in the text when they sum to > 100% due to overlap or non-mutually-exclusive outcomes (e.g. lines 153 – 156). The caption for figure 3 was useful and a similar note should be made in the text where appropriate (e.g. “Many publications used more than one method to monitor contacts, and therefore the numbers of studies exceed 100% for some groups.”)

5. Table 2 also had numerous minor calculation errors, please see individual notes in the PDF and change accordingly. Recommend using more concise language in Tables 2 and 3 (see PDFs for specific suggestions).

6. Figure 3 (and supplemental figure S4): I recommend increasing the contrast between the shades of blue for the dots. It does not stand out very well currently.

7. Figure and table captions/legends should provide more information, in general. The figures and tables should provide enough information to stand on their own (see notes in PDF). The exception is Figure 4, which contains far too much information and is taken verbatim from the text. This should be simplified and redundancy removed.

8. It is already implied that findings in this paper are "agreed upon by the authors" and it’s not necessary to mention this "agreement" several times (e.g. lines 106, 116, 123).

9. There was no problem accessing the raw data provided (Supplementary_File_3)
a. It’s unclear why the first reference says “no author name available.”

10. Supplemental tables and figures do not have titles or captions. Tables S4 and S5 do not match the text of lines 156 – 158 (the text has a denominator of 80 for both diseases, while the total bacterial diseases in Table S4 is 58 and the total viral diseases in Table S5 is 13) and Table S5 has a typo (should read 62% cumulative for foot-and-mouth disease).

11. Manuscript structure and organization were clear and easy to follow; however, it would be helpful to add some additional subheadings to organize the flow of the paper (see PeerJ Literature Review standard sections: https://peerj.com/about/author-instructions/#literature-review-sections)

Experimental design

1. The research question and aims were generally well-defined and clearly stated; however, there are some points of clarification or further discussion needed either in the introduction or discussion.
a. The systemic review question (lines 90-92) includes “risk factors for contacts between wild animals and livestock”, but risk factors were not addressed clearly in this review. Can this be included/addressed or else removed from this sentence?
b. The systemic review question also said, “… contacts between wild animals and livestock or livestock farms worldwide?”. A brief discussion on reporting/ publishing bias is important for any review article. It was useful that you included a geographic breakdown of where articles were published (e.g. Table S3), but it is important to point out that the predominant countries where livestock-wildlife articles were published was the U.S. and U.K., which is hardly representative of the world at large. In order to promote critical thinking and transparency, it recommended that they include the potential bias this introduces into their review in the discussion section. It could be interesting to combine Table S1 (regional distribution) with Figure 2 (yearly distribution) in the same figure with 2 panels, particularly since both space and time are prominent themes of the review.
c. The aims of the review (lines 11-13) included investigating “which livestock-wildlife contacts have been studied and why,” but this did not receive as much attention as the second aim, which was to investigate the observational methods used. More time and thought should go into the discussion of “why” certain livestock-wildlife contacts have been studied. In addition, it’s unclear whether they are referring to the types of contact (e.g. direct vs. indirect) or the species of livestock and wildlife that have been studied.

2. The methods were well described, with sufficient detail and information to replicate. The adherence to PRISMA reporting guidelines was very helpful. Some clarification is needed:
a. Lines 116 – 121: can you clarify whether or not your working/provisional definitions of direct and indirect contact fail to mention space and time considerations? And I’m assuming that these were developed a priori before performing the literature review, but can you confirm?

3. The description of the development of a generic unified framework (lines 131 – 134) in the Methods section is currently quite vague and more detail is warranted. For example, was this framework used to refine your provisional definitions of direct and indirect contact and if so, what is the improved/revised definition? How did you define “meaningful” contacts in each study? In contrast, the results section is overly complicated in its explanation of the framework, both in the text (lines 235 – 244) and Figure 4 (as noted above, the wording is redundant). The way it is currently written is not intuitive and makes the topic more complicated than it needs to be. Also, lines 233 – 235 and 244 – 247 are very similar and these ideas only need to be conveyed once in a single sentence. Some of the scenarios in Figure 4 are somewhat helpful, but I don't understand why scenarios B and D are both considered "near in space" when both involve badgers, but scenario B space is 10-15 m, while scenario D space is >50 m. This seems like a substantial difference in "space."
4. This review article fits within the scope of PeerJ since it is focused on the biological sciences. I commend you for utilizing the most current PRIMSA checklist for literature reviews (and thank you for including it in the supplemental materials). Inclusion and exclusion criteria were clearly defined in the text and Table 1.

Validity of the findings

1. In the abstract (lines 2 –4) and discussion (lines 312 – 314), COVID-19 is mentioned very briefly, although your paper focuses on diseases arising from contact between wildlife and livestock, not humans. It would be more beneficial to highlight emerging infectious zoonotic diseases that impact people directly through livestock that were infected from wildlife (e.g. as you mentioned, Hendra and Nipah are good examples of this). Although COVID-19 is an important emerging infectious zoonotic disease, the most recent evidence of spillover is from bats and potentially other wildlife species in wet markets. I’m not aware of any evidence that livestock played a role in disease transmission. It is misleading to begin your abstract with COVID-19 simply because it's a current public health emergency with international attention, when your paper has a different focus. If you are going to mention it in the discussion, please elaborate more.

2. I was initially concerned that your paper excluded small mammals (rodents, bats) and birds, particularly given their global importance in disease spill-over dynamics. For example, Wiethoelter et al (PNAS, 2015) states, “The bird–poultry interface was the most frequently cited wildlife–livestock interface worldwide.” I was grateful to see this omission partially addressed later in the discussion (lines 329 – 333), but additional explanation would make the paper much stronger. Simply stating that your research does not address avian influenza is insufficient. It is likely that other readers will have similar concerns and will expect more in-depth discussion of this point. I don't think you need to re-do your entire study, but more time must be devoted to justifying why these important animals were omitted, and how your findings may not be reflective of these other important wildlife taxa (e.g. how to interpret your results, recognizing the bias this contributes, etc.). Also, reconsider whether you should use the term “most” wildlife and livestock interfaces throughout the paper because I question whether this is accurate, given your omission of several key taxa.

3. Your discussion section and study conclusions need revisions that include a more in-depth examination of this broad topic and future directions.
a. I think that more persuasive arguments need to be made as to why a unified generic framework is necessary. Lines 252 – 253 give a reason of “wide-ranging definitions describing contacts,” but this is not surprising given the fact that the details or livestock-wildlife contacts are highly context-dependent and variable based on the species of wildlife and livestock (as well as demographics like age and sex), the pathogen (infectious dose, mode of transmission, half-life, etc.), and environmental conditions (temperature, humidity, weather, region, etc.). You do a nice job mentioning some of this in lines 204 – 206, but more is needed. One of the more important points that you make (lines 215 – 216 and 225) is the complete lack of definition of contact, omission of a space or time component, and conflicting or overlapping definitions of direct and indirect contact. These are fundamental problems that make comparisons between studies impossible and necessitates the use of a generic framework like the one you suggest. This needs to be emphasized in the discussion/conclusion of your paper.
b. Lines 22 – 24 mention that this framework “may aid the development of better disease transmission models and improve the design and effectiveness of interventions to reduce or prevent disease transmission.” This sounds great, but is not addressed anywhere else besides your abstract. Further elaboration on this point in your discussion section would be worthwhile.
c. Try to identify unresolved questions, remaining knowledge gaps, and future directions for research pertaining to the livestock-wildlife interface. This will help fit your review into the greater context.

4. I applaud the fact that you mentioned study power and uncertainty in the publications, since this is important for interpretation of findings. You report the number and percentage of studies with low power (lines 147 – 148), but you do not indicate the number or percentage that reported adequate power, or the total number that assessed statistical power (unfortunately, this is commonly omitted from studies, so it would be helpful to know how many even addressed the topic). Can you provide more information here, especially since you mention that power assessment was a criterion for your publication quality score? Also, in lines 322 – 324, you mention low power, but I’m unsure what you meant by “considering the relatively rare nature of certain types of direct contact.” Are you referring to cryptic wildlife species and our inability to capture or document instances of direct contact using current observational methods? Can you clarify and be more specific?

5. Thank you for acknowledging that diseases can also go the other way, from livestock to wildlife. Lines 285 – 287 provide some examples that are much needed. I think more recognition of this bidirectionality aspect is crucial. You mention that there is “little (if any) focus on the value of wildlife,” so I recommend you take this opportunity in the introduction and discussion of your own review paper to highlight the conservation implications of disease spillover into wildlife.

Additional comments

I commend you for your thorough review and adherence to the PRISM reporting guidelines. You have summarized a lot of information in a fairly efficient manner that was clear and easy to follow. You stated how the research fills an identified knowledge gap. Potential weakness of the manuscript include the statistical summaries provided in Table 4 (as I have noted above), the description of the unified generic framework and its utility, and the discussion section, which should be improved upon before Acceptance.

---

## Round 0.2 · accepted · Accept

I am pleased to inform you that your manuscript has been accepted, congratulations and best wishes.

Reviewer 2 ·

Basic reporting

No comments

Experimental design

No comments

Validity of the findings

No comments

Additional comments

I would like to congratulate the authors on the clear effort they have made to improve the manuscript. I really enjoyed reading the document. The ideas are now much clearer and discussion allows understand both the aim of the study avoiding misunderstandings. Congratulations.

·

Basic reporting

No comment.

Experimental design

No comment.

Validity of the findings

No comment.

Additional comments

The authors have done a very nice job incorporating our suggestions and responding to our questions. The tables and figures (especially Table 4) are much improved and the discussion is significantly more thorough and well-organized. Thank you for the opportunity to provide feedback on your manuscript.